# Molecular Genetics and Functional Analysis Implicate *CDKN2BAS1-CDKN2B* Involvement in POAG Pathogenesis

**DOI:** 10.3390/cells9091934

**Published:** 2020-08-21

**Authors:** Sonika Rathi, Ian Danford, Harini V. Gudiseva, Lana Verkuil, Maxwell Pistilli, Sushma Vishwakarma, Inderjeet Kaur, Tarjani Vivek Dave, Joan M. O’Brien, Venkata R. M. Chavali

**Affiliations:** 1Scheie Eye Institute, Department of Ophthalmology, Philadelphia, PA 19104, USA; Sonika.Rathi@pennmedicine.upenn.edu (S.R.); ian.danford@gmail.com (I.D.); gudiseva@pennmedicine.upenn.edu (H.V.G.); lanav@pennmedicine.upenn.edu (L.V.); pistilli@pennmedicine.upenn.edu (M.P.); 2Casey Eye Institute, Oregon Health & Science University, Portland, OR 97239, USA; 3Prof Brien Holden Eye Research Centre, L. V. Prasad Eye Institute, Hyderabad, Telangana 500034, India; svishwakarma17@gmail.com (S.V.); inderjeet@lvpei.org (I.K.); 4Ophthalmic Plastic Surgery Service, Prof Brien Holden Eye Research Centre, L. V. Prasad Eye Institute, Hyderabad, Telangana 500034, India; tarjani@lvpei.org

**Keywords:** *CDKN2B-AS1*, senescence, Primary open-angle glaucoma (POAG), trabecular meshwork cells, African Americans

## Abstract

The genes in the 9p21 locus (*CDKN2B-AS1* & *CDKN2B*) are widely associated with Primary open-angle glaucoma (POAG). However, the functional importance of this locus in POAG pathogenesis is still unexplored. This study investigated the role of *CDKN2BAS1-CDKN2B* axis in POAG. We observed significant association of *CDKN2B-AS1* SNP rs4977756 with POAG and its endophenotypic traits (vertical cup-disc ratio (*p* = 0.033) and central corneal thickness (*p* = 0.008)) by screening African American POAG cases (*n* = 1567) and controls (*n* = 1600). A luciferase reporter assay in Human embryonic kidney 293T (HEK293T) cells revealed that the region surrounding rs4977756 likely serves as a transcriptional repressor. siRNA-mediated knockdown of *CDKN2B-AS1* in HEK293T cells and trabecular meshwork (TM) cells resulted in significantly increased expression of *CDKN2B*, which was also observed in human POAG ocular tissues. Pathway focused qRT-PCR gene expression analysis showed increased cellular senescence, TGFβ signaling and ECM deposition in TM cells after *CDKN2B-AS1* suppression. In conclusion, we report that *CDKN2B-AS1* may act as a regulator, and it could function by modulating the expression of *CDKN2B*. In addition, increase in CDKN2B levels due to *CDKN2B-AS1* suppression may result in the senescence of trabecular meshwork cells leading to POAG pathogenesis.

## 1. Introduction

Primary open-angle glaucoma (POAG) is the most common form of glaucoma, characterized by optic nerve (ON) degeneration and progressive loss of visual field leading to irreversible vision loss. POAG affects over 56 million people worldwide [1] and the number of patients diagnosed with POAG is expected to increase with increasing awareness and improvements to diagnostic technology [2]. Previous studies have revealed a higher prevalence of POAG in African Americans compared to European Americans and other populations globally [3] and in the United States [4,5].

POAG is a complex genetic disorder influenced by the interaction of multiple genes, risk factors and endophenotypic traits such as intraocular pressure (IOP), central corneal thickness (CCT), retinal fiber layer thickness (RNFL), and cup-to-disc ratio (CDR). Genome-wide association studies (GWAS) and candidate gene screening studies have identified the association of many genes including myocilin (*MYOC*), optineurin (*OPTN*), WD repeat domain 36 (*WDR36*), neurotrophic factor 4 (*NTF4*) and ankyrin repeat and SOCS box containing 10 (*ASB10*) with POAG and its endophenotypic traits [6,7,8,9,10,11,12]. Variants in these POAG associated genes may impair the proper functioning of retinal ganglion cells (RGCs) or trabecular meshwork cells (TMs) leading to optic nerve cupping and ocular hypertension [13].

Association studies across different populations have identified the 9p21 locus, consisting of *CDKN2B-AS1*, *CDKN2B*, and *CDKN2A* genes, as associated with POAG and its endophenotypic traits (VCDR and IOP) [14,15,16,17]. SNPs in this locus are also associated with atherosclerosis and cardiovascular diseases [18,19], type 2 diabetes [20], intracranial aneurysm [21], multiple forms of cancer [22] and inflammatory disease [23]. The *CDKN2B-AS1* gene lies in the centromeric region of the 9p21 locus, and encodes a long noncoding RNA also known as *ANRIL*. This noncoding RNA plays a role in epigenetic regulation of gene expression through cis and trans-mechanisms [24]. The *CDKN2A* (cyclin dependent kinase inhibitor 2A) gene lies in the 5′ end of *CDKN2B-AS1* and encodes two proteins, p16INK4A, and p14ARF that differ only in exon 1 due to alternative splicing. *CDKN2B* (cyclin dependent kinase inhibitor 2B) gene lies within exon 1 of *CDKN2B-AS1* in the antisense direction and encodes a p15INK4B protein (Appendix A). These proteins are cyclin-dependent kinase inhibitors involved in cell cycle regulation, DNA damage, apoptosis, senescence, aging, extracellular matrix remodeling, and inflammation. Functional studies on *CDKN2B-AS1* suggested that its function is tissue and cell type specific, and may depend on differential isoform expression of its transcripts. Knockdown of *CDKN2B-AS1* is reported to result in premature senescence of vascular smooth muscle cells [25], and epithelial ovarian cancer cells [26], whereas it reduces or inhibits senescence in endothelial cells [27]. This variability in *CDKN2B-AS1* transcript function highlights the need to studying its role in the ocular cell types to understand its significance in POAG pathogenesis.

Published studies have reported the association of the 9p21 locus with POAG mainly in Caucasian, Asian [17,28] and only in a few Africans populations [29,30,31] (Appendix A) underlining the lack of diversity in previous studies. Moreover, these studies did not show the functional importance of this locus to POAG pathogenesis. In this study, we test the association of the SNP rs4977756 in *CDKN2B-AS1* with POAG and with endophenotypic traits in a large, deeply phenotyped AA cohort with POAG cases and controls. We investigate if the genomic region surrounding this SNP has a regulatory function. We characterize the expression of *CDKN2B-AS1* and its ability to regulate *CDKN2B* and *CDKN2A* transcripts using relevant cell lines including HEK293T cells, human primary TM and human retinal tissues. Gene knockdown studies determine the function of *CDKN2B-AS1* and its role in pathways relevant to POAG pathogenesis.

## 2. Materials and Methods

### 2.1. Enrollment of the Study Subjects

Subjects for this study were part of the Primary Open-Angle African American Glaucoma Genetics (POAAGG) cohort, which is a study investigating the genetic architecture of POAG in the over affected AA population. The POAAGG cohort includes self-identified African American individuals (Black, African decent, or Afro Caribbean), aged 35 years or older, from the Philadelphia region. Eligible patients were enrolled from ophthalmology clinics at the University of Pennsylvania (UPenn) and two external sites in West and North Philadelphia. Our study samples include AAs subjects with POAG (*n* = 1567) as cases and age and ethnicity matched healthy subjects without POAG (*n* = 1600) as controls. The race, age, family history, comorbidities and relevant medical history of all participants was recorded on a pre-designed questionnaire for clinical correlation. This study was approved by the UPenn institutional review board (IRB) and adhered to the Declaration of Helsinki. Subjects with co-morbid eye disease were excluded from the study. The diagnosis of POAG was based on patients having an open iridocorneal angle along with characteristic glaucomatous optic nerve findings corresponding to visual field loss. Blood and saliva samples from the subjects were collected in heparinized vacutainers (BD Biosciences, San Jose, CA, USA) by venipuncture with prior written informed consent and were immediately transferred to the laboratories in ice containers for genetic analysis [32].

### 2.2. Molecular Genetic Analysis

Genomic DNA was isolated from peripheral blood and saliva samples of the study participants using the phenol-chloroform method [33]. The DNA region (801 bp) containing the SNP rs4977756 in the *CDKN2B-AS1* gene was amplified by a polymerase chain reaction (PCR) with gene specific primers followed by bidirectional sequencing (ABI3130xl, Applied Biosystems, Inc., Foster City, CA, USA). The association of genetic variants or SNP rs4977756 with POAG was assessed after determining the SNP status in all the study subjects using appropriate statistical tests. ANOVA analysis was performed on pooled case and control data to determine if the following continuous phenotypic variables related to POAG were associated with variation at this locus: IOP, central corneal thickness (CCT), vertical cup to disc ratio (VCDR), and retinal nerve fiber layer thickness (RNFL). A type 1 error rate of α = 0.05 was chosen to delineate significance for logistic regression and ANOVA analyses.

### 2.3. Generation of Luciferase Reporter Constructs and Luciferase Reporter Assay

Inserts containing the following genotypes at SNP rs4977756 (G/G; A/G; and A/A genotypes) within the *CDKN2B-AS1* region were amplified from genomic DNA isolated from study participants. To generate the A/G genotype from an insert containing the G/G genotype, we used QuikChange XL mutagenesis kit (Agilent Technologies, Santa Clara, CA, USA) as described previously [34]. Amplification of the inserts containing the A/A and G/G genotypes was performed using Phusion High-Fidelity DNA polymerase (NEB) and primers (5′-ACAAGAGCAGGATTGAGTCATGTA-3′ and 5′-TGCATCTTTCTGTCAACTCCACTC-3′) to produce four 2.7 kb DNA constructs surrounding SNP rs4977756 with the above-mentioned genotypes. An *Acc65I* restriction site was added to the sense primer and an *EcoRV* site to the antisense primer. Sequences containing the A/A, G/G, and A/G genotypes were each ligated into a pGL4.11 promoter vector and a pGL4.23 enhancer vector (Promega). The resulting plasmids were transformed into 10-beta Competent *E. coli* (High Efficiency) cells (NEB) and purified using GeneJET Plasmid Midiprep Kit (ThermoFisher). Plasmid sequences were confirmed using Sanger sequencing with Big Dye X Terminator (ThermoFisher).

Using a dual luciferase reporter assay, the enhancer and promoter activity of insert sequences containing A/A, A/G, and G/G were determined by transfecting HEK293T cells in triplicate along with positive and negative control expression vectors (empty pGL4.11 and pGL4.23 vectors). The pGL4.74 renilla luciferase vector (Promega) was co-transfected with experimental plasmids to control for transfection efficiency.

Briefly, HEK293T cells were transfected at approximately 60% confluency with the reporter plasmids using Lipofectamine 3000 (ThermoFisher), according to the manufacturer’s instructions. Following transfection, the cells were incubated at 37 °C with 5%CO_2_ for 48 h. Cells were subsequently lysed, and luciferase activity was measured by performing a Dual-Luciferase assay (Pierce) on transferred lysates in a 96-well white opaque bottomed luminometric plates (Falcon). Measurements obtained for firefly luciferase activity were divided by renilla luciferase activity to control for transfection efficiency, and normalized by the expression from empty vectors [34]. pRLU and empty-vector normalized luciferase expression results were compared by student’s *t*-test for differential luciferase activity.

### 2.4. Cell Lines and Tissues Used for the Study

The HEK293T (human embryonic kidney-CRL3216) cells were obtained commercially from ATCC and primary human trabecular meshwork (hTM) cells (Lot # 3194) from Cell Applications. Characterization of TM cells was done by analyzing the *MYOC* gene expression after inducing dexamethasone (Appendix A) as described previously [34]. All the cell lines used in our study were from passage 4 to passage 8. Retinal tissues were surgically removed during evisceration surgery for POAG patients (40.44 ± 2.3, *n* = 3) while control (76 ± 4.9, *n* = 3) retina tissues were obtained from the donor eyes collected at Ramayamma International eye bank, L. V. Prasad Eye Institute, Hyderabad. The diagnosis of controls samples was confirmed from the history of the patients obtained from the family.

### 2.5. Immunostaining

For immunolocalization of CDKN2B, HEK293T and TM cells grown to 60–70% confluence in chamber slides were fixed with absolute methanol for 15 min in −20 °C and then permeabilized by incubating with 0.5% Triton X in PBS for 5 min at room temperature. Blocking was carried out with 5% normal goat serum in 1% (*w*/*v*) BSA in 1X PBS for 60 min at room temperature. Thereafter, the slides were incubated for 60 min with rabbit Anti-p15INK4b antibody (ab53034) diluted to 1:100 in 0.1% BSA, followed by three washings with 1X PBS. Further incubation was carried out with Alexa 594 anti-mouse secondary antibody for one hour. Cells were washed in 1X PBS and mounted with prolong gold mounting medium with DAPI for nuclei staining. Images were captured by Olympus FV1000 Confocal microscope with the use of appropriate filters and lasers [35].

### 2.6. CDKN2B-AS1 siRNA Treatment

TM cells were grown to approximately 80% confluence in 6-well plates. The cells were transfected with 10 nM of pooled *CDKN2B-AS1* siRNA (Dharmacon, Lafayette, CO, USA; Cat# R-188105-00-0005) and 10 nM of non-targeting control pool siRNA (Dharmacon Research; D-001320-10) using Lipofectamine RNAiMAX (Invitrogen, Carlsbad, CA, USA). After 72 h, RNA was extracted from the transfected TMs and converted to cDNA using Superscript III reverse transcription kit (Life Technologies, Carlsbad, CA, USA) following manufacturer’s instructions. Quantitative real-time PCR with gene specific primers was done following the methodology described below. These experiments were done in triplicate.

### 2.7. β-Galactosidase Assay

Senescence in TM cells was determined by using β-galactosidase staining kit (Cell Signaling Technology, Danvers, MA, USA). Briefly, TM cells at 80–90% confluence were washed twice with 1X PBS to remove growth media and fixed in a fixative solution for 10–15 min at room temperature. Cells were washed twice with 1X PBS and incubated in dark overnight at 37 °C with fresh β-galactosidase staining solution. Stained cells were imaged under a light microscope and examined for the development of blue color.

### 2.8. Treatment of Cells with TGF-β1 and TGF-β2

Primary TM cells (Passage 6) at 70–80% confluence were treated with 2.0 ng/mL activated rhTGF-β1 and rhTGF-β2 (R&D Systems, Minneapolis, MN, USA) for 72 h. The medium was replaced every 24 h with fresh medium containing recombinant human TGF-β1 and TGF-β2 (at a final concentration of 2 ng/mL) and incubation was continued for a total of 72 h. Cells treated with vehicle, 4 mM HCl containing 1% BSA was used as a control.

### 2.9. Semi-Quantitative PCR

The RNA from untreated and treated TMs, HEK293 cells (using *CDKN2B-AS1* siRNA and TGFβ induction) and POAG and normal eye tissues was extracted by the Trizol method (25). The cDNA was prepared using Superscript III One-Step RT-PCR kit (Invitrogen, Carlsbad, CA, USA). The RNA isolated from retinal tissues (POAG and normal retinal tissues) was converted to cDNA using verso cDNA synthesis kit (Thermo Scientific™, catalog no. AB1453B). A 10 μL reaction mixture was made using iTaq™ Universal SYBR^®^ Green Supermix (BIO-RAD, Cat no. 38220090). Quantitative PCR was carried out using gene specific primers (Appendix A) for *CDKN2B-AS1*, *CDKN2B*, *CDKN2A*, *P14ARF*, β-actin and *GAPDH*. Pathway focused gene expression analysis was done using RT2 profiler PCR array (Human TGFβ/BMP signaling, Cat# PAHS-035Y, Human Extracellular Matrix and Adhesion Molecules Cat#PAHS-013Z and Human Cellular Senescence Cat# PAHS-050Z) following manufacturer’s instruction.

## 3. Results

### 3.1. Genotype-Phenotype Association Studies for SNP rs4977756

The candidate SNP study for rs4977756 revealed that the A allele reached nominal statistical significance, *p* = 0.042, according to logistic regression analysis controlling for age and sex (Table 1). Additionally, analysis of the odds ratios (OR) for the genotypes suggested a recessive model of POAG risk for the risk allele A: OR [AG] = 1.02 (0.80, 1.30) and OR [AA] = 1.21 (0.94, 1.55). Therefore, we also performed logistic regression to determine the risk of POAG for patients harboring zero or one risk alleles versus those harboring two risk alleles. The association became more significant (*p* = 0.027), with an OR of 1.19 for homozygotes of the A allele (Table 2).

Analyses of the association between rs4977756 and quantitative POAG risk factors also yielded significant associations at α = 0.05. Prior to applying the recessive model of POAG risk for the A allele, ANOVA analysis revealed that the SNP rs4977756 was associated with CCT in a combined case-control analysis pool (*p* = 0.0299). Specifically, the AA genotype was associated with a thinner CCT than the GG and AG genotypes. When all three genotypes were considered in the ANOVA analysis, no other quantitative trait reached the threshold of statistical significance (Table 3).

However, when the recessive model for the A risk allele was used, we found that VCDR was also associated with the rs4977756 locus in our study population (*p* = 0.0326). Additionally, the association between rs4977756 and CCT became stronger (*p* = 0.0081). This analysis suggests that individuals with the AA genotype have a significantly lower CCT measurements and greater VCDR measurements compared to individuals with the GG or AG genotype at rs4977756 (Table 4). We also performed multivariable linear regression to test the associations between rs4977756 (recessive model) and CCT and VCDR, controlling for age, sex, and disease status. The relationship between CCT and rs4977756 remained significant (*p* = 0.011), however the relationship between VCDR and rs4977756 did not reach significance (*p* = 0.261) (Table 4).

### 3.2. Potential Regulatory Role of SNP rs4977756

We observed an increased risk for POAG in patients with a homozygous A allele for SNP rs4977756. These findings were consistent with published studies [14,17]. In contrast, patients with the homozygous G allele have been previously shown to have increased risk of high-grade gliomas (excessive glial cell division and multiplication) [36]. These results suggest that this SNP region may have a regulatory role and can alter the expression of *CDKN2B-AS1* transcripts. Hence, we performed a luciferase reporter assay to understand the regulatory nature of this region. Analysis of the luciferase expression revealed that the region cloned upstream to the luciferase gene in the pGL4.11 promoter vector resulted in extremely low mean luciferase expression when compared to renilla luciferase expression and empty vector. This suggests the 2.4 kb DNA insert containing rs4977756 is unlikely to be a promoter (Figure 1).

When the genomic DNA with varying genotypes for rs4977756 were cloned downstream to a minimal promoter into the pGL4.23 repressor/enhancer construct, they exhibited a significant reduction in the mean luciferase expression when compared to the empty pGL4.23 vector, indicating that these regions may act as a repressor of luciferase gene expression (Figure 1). Specific genotypes within these inserts were associated with differing mean luciferase expression levels. There was no significant difference in mean luciferase expression between pGL4.23 inserts containing G/G and A/G genotypes. However, the A/A genotype was associated with significantly higher mean luciferase expression than the A/G and G/G genotype (Figure 1).

### 3.3. Transcript Expression of CDKN2B-AS1 Locus in POAG Retina, TM and HEK293T Cells

An increased risk for POAG in patients with SNP rs4977756 and repressor activity of the genomic region surrounding this SNP by luciferase reporter assays suggests that this region may be functionally relevant to glaucoma pathogenesis. We observed a significant increase in the expression of *CDKN2A* (*p* = 0.011) and *CDKN2B* (*p* = 0.002) transcripts in POAG retina when compared to normal/control retina, further supporting their involvement in POAG pathogenesis (Figure 2a). We also determined the endogenous expression of *CDKN2B-AS1* and its nearby genes *CDKN2B* and *CDKN2A* in HEK293T and TM cells. Expression of *CDKN2A*, *CDKN2B* and *CDKN2BAS1* transcripts are quite similar in HEK293T and TM cells. However, there was higher expression of *CDKN2BAS1* in HEK 293T cells as compared to TM cells. Interestingly, *CDKN2A* transcript was abundantly expressed when compared to *CDKN2B* in HEK293T and TM cells (Figure 2b).

### 3.4. CDKN2B-AS1 Regulates the Expression of CDKN2B and CDKN2A Transcripts

Published studies on peripheral blood cells in diabetes, cardiac diseases and cancers showed the regulatory effect of *CDKN2B-AS1* on the expression of *CDKN2A* and *CDKN2B* transcripts [37]. However, this regulatory potential of *CDKN2B-AS1* has not been tested in ocular cell lines and tissues. Hence, we used siRNA knockdown studies to determine if the *CDKN2B-AS1* knockdown affects cis-gene expression. Real-time qRT-PCR analysis showed efficient knockdown of *CDKN2B-AS1* by the pooled 10 nM *CDKN2B-AS1* siRNAs in both TM and HEK293T cells when compared to cells transfected with non-targeting control siRNA (Figure 3). Knockdown of *CDKN2B-AS1* resulted in significant increase in *CDKN2B* expression in HEK293T cells (Figure 3a,c). A marked increase in *CDKN2B* expression was also observed in TM cells after 10 nM *CDKN2B-AS1* siRNA treatment as assessed by immunocytochemistry and RT-PCR (Figure 3b,c). No significant change in *CDKN2A* and *p14ARF* expression was observed in TMs and HEK293T cells after *CDKN2B-AS1* suppression. Taken together, *CDKN2B-AS1* siRNA knockdown has similar effects on the expression of *CDKN2A*, *CDKN2B* and *p14ARF* genes in TM and HEK293T cells, but with varying intensity.

### 3.5. Role of CDKN2B-AS1 in Causing Cellular Senescence and ECM Homeostasis

Accumulation of cells expressing cellular senescence markers were found to be associated with several age-related diseases including POAG. Published studies reveal the potential role of accumulation of senescent TM cells in the pathophysiology of the aqueous humor outflow pathway function [38]. In vitro experiments showed the effect of *CDKN2B-AS1* knockdown in increased apoptosis and senescence, while its overexpression reversed the phenotype in cancer cells [39]. In the present study, increased expression of senescence pathway genes (Figure 4a) was observed in TM cells after *CDKN2B-AS1* siRNA treatment when compared to the TM cells treated with non-targeting control siRNA using RT2 profiler array. *CDKN2B-AS1* suppression resulted in a significant increase in senescence initiators pathways including interferon signaling (*CDKN1A*, *CDKN1B*, and *ALDH1A3*), insulin growth factor (*IGFBP5*, *IGF1R*, *IGFBP3*) and p53/pRB signaling genes (*ING1*, *ID1*, *PLAU*, *CDKN1C*, *SPARC* and *CDKN2B*) (Figure 4b). Significantly, increased expression of cell adhesion molecules in TM cells (*CD44*, *ICAM1*, *CTNNA1*, *CTNND1*, *LAMA1*, *LAMC1*, *SPARC*, *ITGA3*, *ITGA5* and *ITGA8)* after *CDKN2B-AS1* siRNA treatment (Figure 4c) supported the presence of senescent environment in TM cells [37,38,39,40,41,42]. Presence of increased number of SA-β-Gal positive TM cells after *CDKN2B-AS1* treatment, when compared to non-targeting control siRNA, confirmed the senescent phenotype. This data further defines the role of *CDKN2B-AS1* in regulating senescence (Figure 4d).

In addition, senescent cells altered the extracellular matrix (ECM) composition by increased expression and degradation of extracellular matrix metalloproteinase, leading to the accumulation of ECM products that may have deleterious effects in tissue physiology. Moreover, stiffening of TM cells is the hallmark of glaucoma and is believed to be induced by ECM deposition. Significant increased expression of ECM molecules *ECM1*, *COL7A1*, *COL12A1*, *COL15A1*, *COL16A1*, *COL1A1*, and *COL6A2* along with MMPs suggests an accumulation of ECM products in TM cells after *CDKN2B-AS1* suppression (Figure 4e). Increase in senescent cells and ECM deposition may affect the TM cell integrity resulting in functional alteration and increased outflow resistance, which may lead to RGC death, thereby contributing to the development or progression of POAG.

### 3.6. TGF-β Signaling in Senescent-Inflammatory Phenotype

The TGFβ signaling pathway is of particular interest in glaucoma, since functional studies have shown the role of elevated TGFβ1 signaling in inflammation, senescence and ECM deposition in TM cells, lamina cribrosa (LC), and aqueous humor of glaucomatous eyes [13,40,41]. Knockdown of *CDKN2B-AS1* increased the expression of p15 (INK4b) and transforming growth factor β1 (TGFβ1) resulting in the inhibition of cellular proliferation in human esophageal squamous cell carcinoma [42]. This study suggested the role of TGFβ1 signaling in *CDKN2B-AS1* mediated regulation in cell proliferation. Therefore, to understand the role of *TGFβ1-CDKN2B-CDKN2B-AS1* axis in glaucoma pathogenesis, we investigated the effect of *CDKN2B-AS1* knockdown and TGFβ1/TGF β2induction on this axis. Knockdown of *CDKN2B-AS1* transcript resulted in an increase of *TGFβ1* and *CDKN2B* transcript expression in TM cells (Figure 5a). Quantitative gene expression profiles showed a significant increase in TGFβ signaling genes like *CDKN1A*, *IGFBP3*, *MYC*, *ID1*, *FAS* and *UBASH3B* involved in cellular senescence; *BMP1*, *COL1A1*, *COL1A2*, *TGFBI* involved in ECM deposition and other signaling genes *SMURF1* and *HERPUD1* in TM cells after *CDKN2B-AS1* suppression (Figure 5a). Downregulation of TGFβ signaling regulators/inhibitors, i.e., *NDR1*, *BMB1*, *PLAU*, *SERPINE1* and *THBS1* (Figure 5a), further strengthened our hypothesis that increased TGFβ signaling mediates senescence and fibrosis after *CDKN2B-AS1* suppression.

Furthermore, induction of TM cells with TGFβ1 and TGFβ2 resulted in a significant increase in *CDKN2B* along with a slight increase in *CDKN2B-AS1* (Figure 5b,c). We observed a significant increase in inflammatory cytokines IL1a and IL6 and increase in TNFα when compared to vehicle induced TM cells, suggesting the role TGFβ mediated inflammation in TM cells (Figure 5d). Thus, our results demonstrate the role of increased TGFβ signaling in cellular senescence, fibrosis and inflammation in TM cells. Although, TGFβ may not be in tight regulation with *CDKN2B-AS1*, our results show that it plays a role in *CDKN2B-CDKN2B-AS1* mediated cellular senescence and inflammation.

## 4. Discussion

This study demonstrated that the *CDKN2B-AS1* involved in mediating senescence, inflammation, and ECM accumulation and may play an important role in the development of POAG. We showed that the intronic region of *CDKN2B-AS1* harboring the allele A at SNP rs4977756 was associated with an increased risk for POAG in AAs. Our results agree with findings reported in populations of European descent [14,28], but to the best of our knowledge, this SNP has not previously been associated with POAG in an AA cohort. Moreover, previous studies on African populations have identified an association of SNPs in the 9p21 locus with POAG (although reported SNPs differed) [31] suggesting that this locus is driving risk of POAG in AAs as in other populations. However, it is important to consider that, unlike Europeans, lack of a larger sample size could be a reason for nominal associations of rs4977756 with POAG in Africans and African descent samples [31].

It was interesting to note that the recessive model at rs4977756 resulted in stronger associations between POAG risk factors (CCT and VCDR). The 9p21 locus is known to be associated with normal tension glaucoma and various optic disc parameters, including CDR, but is not a genetic predictor of CCT [28,43]. CCT is an independent risk factor for the development of POAG, such that patients with a thinner CCT at a relatively elevated risk for developing POAG, regardless of their IOP. This correlation led to the hypothesis that a thinner CCT may be a biomarker of structural or ECM abnormalities found within the TM or ONH. This idea was further supported by our functional studies in TMs where knockdown of *CDKN2B-AS1* resulted in significant disruption in ECM homeostasis. Although we did not quantify CCT thickness in AAs, it was interesting that SNP rs4977756 in *CDKN2B-AS1* was strongly associated with CCT and VCDR, but not IOP.

It is hypothesized that the intronic region containing rs4977756 may possess response elements that affect the expression of *CDKN2B-AS1* and its sense transcripts [31]. Interestingly, eQTL studies in lymphoblastic cell lines have shown that rs4977756 genotypes significantly affect the expression of the non-adjacent gene, dehydrogenase/reductase 9 (*DHRS9*) (i.e., trans-regulation) [24]. *DHRS9* participates in the metabolism of all-*trans* retinoic acid in neurons and astrocytes, and can also serve as a nuclear transcriptional repressor [44]. Our luciferase assay results demonstrated that the intronic region surrounding rs4977756 contains regulatory response elements more specifically, likely to serve as transcriptional repressors. Our results also suggested that specific allele combinations at rs4977756 may fine-tune the repressor activity in this region, as inserts containing the A/G rs4977756 genotype exhibited less luciferase expression that the G/G and A/A genotypes. Further, we used the bioinformatics tool MATCH to predict whether the short segment containing rs4977756 loci might represent known transcription factor binding motifs [45]. According to MATCH, the COMP1 transcription factor was repeatedly implicated to bind rs4977756 loci. COMP1 is relatively understudied, but it is thought to regulate inflammatory responses [45]. These transcription binding sites may affect inflammatory responses and senescence that correspond well to the senescent-inflammatory phenotype of POAG previously described [13].

Disease associated variants have been reported to result in differential transcript expression of *CDKN2A*, *CDKN2B* and *CDKN2B-AS1* [46,47]. Cardiac artery disease (CAD) associated risk-SNPs increased *CDKN2B-AS1* isoform expression in human endothelial cell lines, macrophage cell cultures and coronary smooth muscle cells [48], while a reduction of *CDKN2B-AS1* and *CDKN2A* expression increased *CDKN2B* expression in peripheral blood cells [37]. The present study is an attempt to test the expression of *CDKN2B-AS1*, *CDKN2B* and *CDKN2A/ARF* in TM cells, which are one of the primary cells affected in POAG, and to correlate their functions with downstream pathway regulation. So far three novel *CDKN2B-AS1* splice variants have been reported in the human retina [14] and (in Ensemble genome browser), 28 *CDKN2B-AS1* splice variants are reported, with exons 1 and 6 coinciding in most of them. Therefore, a pool of four siRNAs targeting exon 1, 2 and 6 were used for *CDKN2B-AS1* knockdown studies in HEK 293T cells and TMs, which resulted in the efficient suppression of *CDKN2B-AS1* expression.

We observed that the suppression of *CDKN2B-AS1* resulted in the upregulation of *CDKN2B* in TMs and HEK292T cells, which are in accordance with previous reports in vascular smooth muscle cells [49] and cancer cell lines [50]. We did not observe any significant change in the *CDKN2A* or *p14ARF* expression following knockdown of *CDKN2B-AS1* transcripts. These findings are consistent with Kotake et al. study suggesting the involvement of polycomb repression complex mechanism in the regulatory role of *CDKN2B-AS1* in an ocular cell line [50]. This mechanism postulated that *CDKN2B-AS1* serves as a scaffold for the chromatin modifying complexes PRC1 and PRC2, mediating the repression in *cis* of the *INK4b*-*ARF*-*INK4a* locus [50,51].

It is well established that *CDKN2B-AS1* plays a regulatory role in senescence and apoptosis through several pathways involved in the cell cycle including DNA damage, apoptosis and inducing a senescence associated secretory phenotype [52]. Signaling pathways, i.e., retinoblastoma protein (pRB) and protein 53 (p53), regulate cell division via CDKN2A and CDKN2B resulting in cellular senescence. Increased expression of p53/p21 signaling and pRB/p16 has been reported to induce premature senescence due to down regulation of *CDKN2B-AS*1 [53]. The *CDKN2B-AS1* knockdown studies in TM cells may have resulted in premature senescence due to increased gene expression in the above pathways (Figure 4). *CDKN2B-AS1* also influences cell growth by repression of the TGFβ/Smad signaling pathway [42]. In the present study, *CDKN2B-AS1* knock down resulted in the increase in TGFβ1 signaling, mediating increase in expression of cellular senescence genes, ECM genes and decrease in expression of TGFβ regulators and inhibitors. The TGFβ induction in TM cells did not correlate directly with *CDKN2B-AS1* mediated cellular senescence, suggesting that TGFβ signaling in cell regulation relies upon a balance of the signaling inputs from other genes, growth factors and cytokines [54]. In POAG, it is most likely that genetic or aging related gene expression changes to *CDKN2B-AS1-CDKN2B* axis may lead to cellular senescence associated alteration in secretome and cytokines, which leads to increase TGFβ1 signaling mediated senescent-inflammatory phenotype or vice versa (Figure 6).

In conclusion, current findings appear to support that at least some POAG associated SNPs (and unassociated SNPs in relative LD) in 9p21 locus may have regulatory roles that could potentially translate to altered expression of genes in this locus in a cell specific and tissue specific manner. Significant increased expression of *CDKN2A* and *CDKN2B* was observed in POAG retinal tissues, further signifying the involvement of this locus in POAG pathogenesis. Our data further suggests that the complex interplay between the genes/molecules regulated by *CDKN2B-AS1-CDKN2B* axis (9p21 region) in TM cells may play an important role in inflammation, senescence and tissue remodeling (Figure 6). Further experimentation is warranted in RGCs and POAG tissues to better elucidate the exact nature of the interactions between genes in this locus that may lead to RGC death in POAG. Understanding the ocular cell type specific expression and regulation of *CDKN2B-AS1-CDKN2B* axis may be the first step towards the understanding of this complex mechanism in POAG pathogenesis. Much remains to be learned about the *CDKN2BAS1-CDKN2B* transcript/isoform expression in ocular cell types and how these different isoforms influence tissue homeostasis to modulate disease risk. A better understanding of these complex mechanisms could have a profound translational impact, in slowing the development of POAG by intervening in neuro-inflammation, tissue remodeling, and cell death.

## Figures and Tables

**Figure 1 cells-09-01934-f001:**
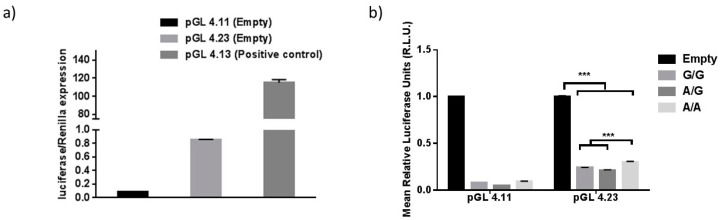
Luciferase Expression results for rs4977756 Genotypes. (**a**) Luciferase expression in promoter construct pGL4.11 (empty), pGL4.23 (empty) and pGL4.13 (SV40 promoter). (**b**). Mean pRLU-Normalized Luciferase Expression for rs4977756 genotypes. Mean luciferase expression from promoter construct vector pGL4.11 was very poor for all inserts. Each insert in the pGL4.23 response-element construct was associated with a reduction in mean luciferase expression compared to the empty pGL4.23 vector. All three *t*-tests comparing mean relative luciferase expression between the empty pGL4.23 vector and vectors containing inserts of varying genotypes (denoted with ***) were statistically significant with *p*-values < 0.0001.

**Figure 2 cells-09-01934-f002:**
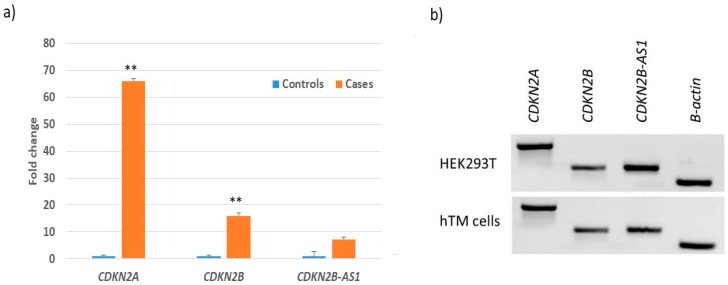
Expression of *CDKN2A*, *CDKN2B* and *CDKN2B-AS1* transcripts in ocular tissue and cells in (**a**) expression in POAG (*n* = 3) and control retina (*n* = 3) and (**b**) human trabecular meshwork (hTM) and Human embryonic kidney 293T (HEK293T) cell lines. ‘**’ signifies *p* < 0.005.

**Figure 3 cells-09-01934-f003:**
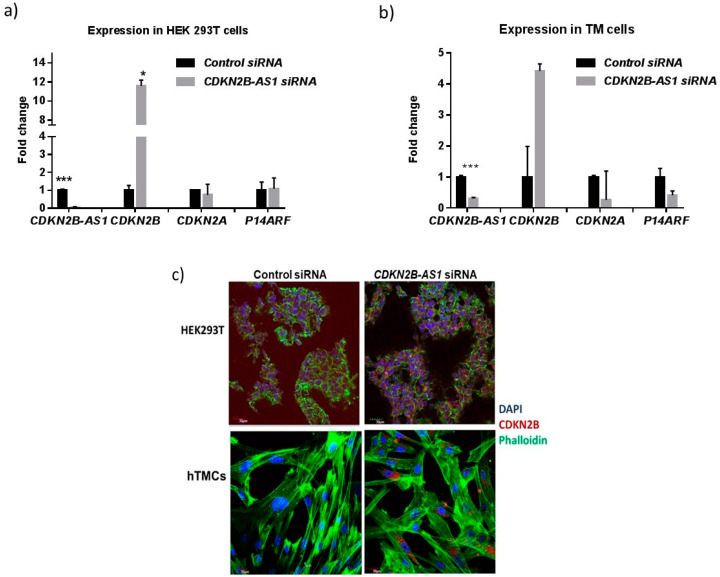
*CDKN2B-AS1* siRNA mediated knockdown in HEK293T and trabecular meshwork (TM) cells. (**a**,**b**) Relative quantification of *CDKN2B-AS1*, *CDKN2B*, *CDKN2A* and *P14ARF* transcript in HEK293T and TM cells after *CDKN2B-AS1 siRNA* (10 nm) treatment when compared to control siRNA (10 nm) treatment for 72 h; (**c**) Representative cultures established after *CDKN2B-AS1* and control siRNA treatment demonstrated increased CDKN2B (red) detection in phalloidin (green) stained HEK293T and TM cells. (Scale bar represents 30 µm; *** denotes *p* < 0.001; * denotes *p* < 0.01, Bar graphs illustrate fold change (2^−ΔΔ^*^C^*^T^) ± SEM).

**Figure 4 cells-09-01934-f004:**
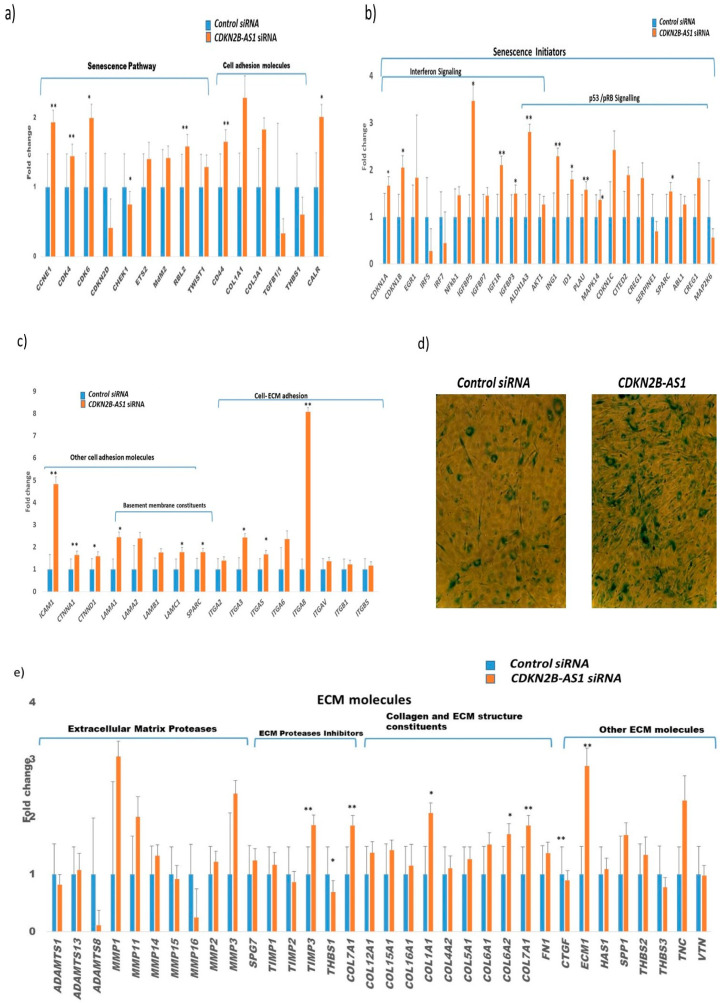
Cellular senescence in TM cells after suppression of *CDKN2B-AS1*. (**a**–**c**,**e**) The human cellular senescence and extracellular matrix and adhesion molecules RT^2^ Profiler PCR arrays showed differential gene expression profile in TM cells after 10 nM *CDKN2B-AS1* siRNA and control siRNA treatment for 72 h. Bar graphs illustrate fold change (2^−ΔΔ^*^C^*^T^) ± SEM. Asterisks indicate significance of difference from controls ** *p* < 0.05, * *p* < 0.005, (**d**) Senescence-related β-galactosidase staining of TM cells after 10 nM *CDKN2B-AS1* siRNA and control siRNA treatment. Blue color stained cells are β-galactosidase positive cells indicates senescence. Magnification 10×.

**Figure 5 cells-09-01934-f005:**
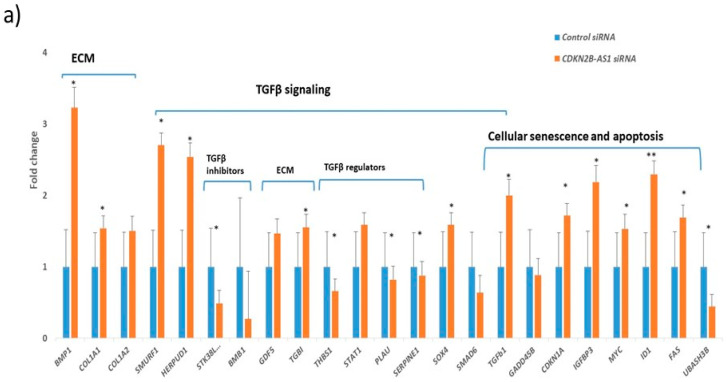
*CDKN2B-AS1* mediated senescence and inflammatory cytokine secretion via TGFβ signaling. (**a**) The human TGFβ BMP Signaling Pathway RT2 Profiler PCR arrays showed differential gene expression profile in TM cells after 10 nM *CDKN2B-AS1* siRNA and control siRNA treatment for 72 h. (**b**,**d**) TM cells were treated with vehicle or 2.0 ng/mL activated rhTGFβ1 for 3 days. (**c**) TM cells were treated with vehicle or 2.0 ng/mL activated rhTGFβ2 for 3 days. *β-actin* normalized transcript levels for *CDKN2B*, *CDKN2B-AS1*, inflammatory cytokines *(IL1a*, *IL1b*, *Il6*, *IL8* and *TNFa)* in TGFβ induced hTM cells relative to control as detected by qPCR. Bar graphs illustrate fold change (2^−ΔΔ^*^C^*^T^) ± SEM. Asterisks indicate significance of difference from controls; *** *p* < 0.05, ** *p* < 0.005, * *p* < 0.005.

**Figure 6 cells-09-01934-f006:**
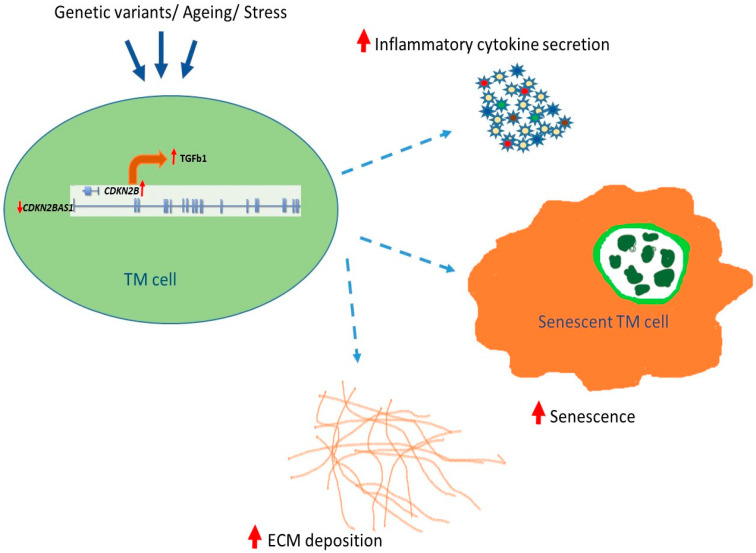
Schematic of the proposed mechanism of *CDKN2B-AS1* knockdown mediated senescence- inflammatory phenotype in TM cells. Genetic variants, aging and oxidative or other stress could result in the downregulation of *CDKN2B-AS1*, resulting in increased expression of *CDKN2B* (cell cycle arrest gene). *CDKN2B* directly or via TGFβ signaling increases ECM deposition, cellular senescence and secretion of inflammatory cytokines in TM cells. Together, these will contribute to increased outflow resistance and TM cell death leading to POAG pathogenesis.

**Table 1 cells-09-01934-t001:** Association between rs4977756 genotypes and Primary open-angle glaucoma (POAG) disease status (case vs. control). Analyses controlled for age and sex.

Locus | Genotype		Case (Frequency)	Control (Frequency)	Total (Frequency)	Age and Sex Adjusted Odds Ratio (95% CI)
**rs4977756**	GG	181 (11.6%)	199 (12.4%)	380 (12.0%)	1
	AG	691 (44.1%)	753 (47.1%)	1444 (45.6%)	1.02 (0.80, 1.30)
	AA	695 (44.4%)	648 (40.5%)	1343 (42.4%)	1.21 (0.94, 1.55)
	Total	1567 (49.5%)	1600 (50.5%)	3167	

Age and Sex Adjusted (Logistic Regression) *p*-value: 0.042 *. * denotes a significant result for α = 0.05.

**Table 2 cells-09-01934-t002:** Association between rs4977756 genotypes and POAG disease status (case vs. control), according to a recessive model of POAG risk. Analyses controlled for age and sex. * denotes a significant result for α = 0.05.

Locus | # of Risk Alleles		Case (Frequency)	Control (Frequency)	Total (Frequency)	Age and Sex Adjusted Odds Ratio (95% CI)
**rs4977756**	0-1 Risk Alleles (GG or AG)	872 (55.7%)	952 (59.5%)	1824 (57.6%)	1
	2 Risk Alleles (AA)	695 (44.3%)	648 (40.5%)	1343 (42.4%)	1.19 (1.02, 1.39)
	Total	1567 (49.5%)	1600 (50.5%)	3167	

Age and Sex Adjusted (Logistic Regression) *p*-value: 0.027 *.

**Table 3 cells-09-01934-t003:** Relationship between rs4977756 genotypes and quantitative POAG risk factors. * denotes a statistically significant result (α = 0.05).

rs4977756					
POAG Risk Factor	ANOVA *p*-Value	GG Mean (Std Dev)	AG Mean (Std Dev)	AA Mean (Std Dev)	All Genotypes
Mean CCT	0.03 *	535.80 (39.50)	535.39 (41.52)	530.28 (38.32)	533.22 (39.98)
Mean IOP	0.65	15.83 (4.57)	16.03 (3.82)	16.06 (4.07)	16.02 (4.02)
Mean VCDR	0.1	0.54 (0.25)	0.54 (0.25)	0.56 (0.25)	0.55 (0.25)
Mean RNFL	0.69	77.47 (13.37)	76.42 (15.19)	76.21 (15.33)	76.45 (15.04)

**Table 4 cells-09-01934-t004:** Relationship between rs4977756 recessive model and quantitative POAG risk factors. The bottom panel includes the p-values for the follow-up linear regression analyses for central corneal thickness (CCT) and vertical cup to disc ratio (VCDR) controlling for age, sex, and disease status. * denotes a statistically significant result (α = 0.05).

rs4977756 (0-1 vs. 2 Risk Alleles)				
POAG Risk Factor	ANOVA *p*-Value	0-1 Risk Alleles Mean (std dev)	2 Risk Alleles Mean (std dev)	All Genotypes Mean (std dev)
Mean CCT	0.008 *	535.48 (41)	530.28 (38)	533.22 (39)
Mean IOP	0.65	15.99 (3.99)	16.06 (4.07)	16.02 (4.02)
Mean VCDR	0.033 *	0.54 (0.25)	0.56 (0.25)	0.55 (0.25)
Mean RNFL	0.64	76.64 (14.8)	76.21 (15.3)	76.45 (15.)

Mean CCT Age, Sex, and Disease Status Adjusted (Linear Regression) *p*-value: 0.011 *. Mean VCDR Age, Sex, and Disease Status Adjusted (Linear Regression) *p*-value: 0.261.

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
