# Peer review of "Molecular Genetics and Functional Analysis Implicate CDKN2BAS1-CDKN2B Involvement in POAG Pathogenesis"

_cells, 2020, doi:10.3390/cells9091934_

Round 1

Reviewer 1 Report

This is an exciting and thorough study on the molecular genetics of the CDKN2BAS1-CDKN2B-CDKN2A locus that has previously been implicated in various diseases; the study also describes functional analysis with implications in POAG susceptibility and pathogenesis. Below are suggestions that I believe will strengthen this paper:

Introduction:

Please elaborate on CDKN2BAS1 vs CDKN2B vs CDKN2A - a figure that shows the complexity of this gene region would be helpful. It would also be useful to explain why the two SNPs (rs4977756 and rs1412831) were chosen, specifically. Further, the allele frequencies in diverse populations of the SNPs of interest, and others in this locus implicated in prior studies along with the diseases in which they are implicated, would be useful to readers displayed as a table or figure to de-convolute this dense information.

Please be clear about the populations that prior studies have been performed in. The authors should emphasize the lack of diversity in many of the prior studies.

It is important to be aware of and present appropriately the differences and use of race, ethnicity, and ancestry. As genetic ancestry evaluation is not mentioned in this paper for characterization of African Americans, it would be helpful to know if the subjects were self-identified, third-party identified, or something else. 

Please be clear whether variants in genes of interest have associated with POAG endophenotypes in AAs previously. 

Methods:

Sections 2.1 and 2.2 mention blood and saliva as sources of DNA; but section 2.2 does not mention any statistical adjustment for these differences in DNA source - please comment. 

Section 2.3: With regard to the age of the POAG patients from which retinal tissues were surgically removed, is there any statistical adjustment made or necessary for the difference in age between tissue sources from cases and controls?

Results:

Section 3.1: last sentence of paragraph 1 - is this consistent with prior knowledge? 

Section 3.2: second sentence "These findings were consistent with our results..." - what results is this referring to?

Section 3.3: first sentence - please reword, this is unclear.

Author Response

Response to Reviewers (Cells-853263-1):

Reviewer 1:

Point 1- Reviewer's comment: Please elaborate on CDKN2BAS1 vs CDKN2B vs CDKN2A - a figure that shows the complexity of this gene region would be helpful. It would also be useful to explain why the two SNPs (rs4977756 and rs1412831) were chosen, specifically.

Response: As per reviewer’s suggestion, we provided a schematic showing the 9p21 locus with genes CDKN2B-AS1, CDKN2B and CDKN2A in Supplementary Figure 1 (lines 489-492 and 56 in the manuscript). Both CDKN2A (p16ink4a) and p14ARF share exon 2 and 3 but differ in exons 1 as a result of alternative splicing. The location of the SNP rs4977756 discussed in the current study is marked in Supplementary Figure 1. Genome-wide association studies in several populations to date implicated stronger association of SNP rs4977756 with POAG [1-4] among other SNPs. In this study, we attempted to functionally investigate the regulatory role of SNP rs4977756 by performing luciferase assay and characterize the function of this locus. We sequenced approximately 801bp of genomic region containing SNP rs4977756 by PCR amplification followed by Sanger sequencing. The SNP rs1412831 is located 6bp upstream to rs4977756 and is a sequencing bycatch. This SNP was reported to be correlated with a GWAS associated SNP rs3217989 that protects against Coronary Artery Disease in African Americans [5]. However, this SNP was never implicated in any POAG GWAS studies to date. Due to its proximity to the known POAG SNP rs4977756, we independently tested its association with POAG case/control status and with endophenotypes (data not shown) and did not find any genomic association. We also tested the regulatory nature of SNP rs1412831 by performing luciferase assays. Based on the above findings, we decided to remove its reference in our revised manuscript to improve the clarity of our results and avoid any confusion.

Supplementary Figure 1: Schematic representation showing CDKN2B-AS1, CDKN2B and CDKN2A gene and location of rs4977756 (indicated by arrowhead) within CDKN2B-AS1 gene..

Point 2-Reviewer’s comment: Further, the allele frequencies in diverse populations of the SNPs of interest, and others in this locus implicated in prior studies along with the diseases in which they are implicated would be useful to readers displayed as a table or figure to de-convolute this dense information.

Response: We thank the reviewer for this suggestion. In the revised manuscript in lines 493-494 and lines 67, we included a new Supplementary Table S2, containing SNPs of interest, and other important SNPs associated with POAG in 9p21 locus with allele frequencies in diverse populations.

Supplementary Table S2: Comprehensive list SNPs associated with POAG in 9p21 locus in diverse populations.

SNP ID

Gene

Population (Cases/

controls)

Association with Phenotype/Traits

Associated allele

MAF in cases/controls

References

rs1063192

CDKN2B

African Caribbean (272/165)

 POAG

C

0.039/0.094

 [3]

American  (539/336)

POAG, VCDR

G

0.342/0.417

[6]

Chinese (1157/934)

POAG

C

0.184/0.204

[2]

Australian  (326/883)

Advanced OAG, VCDR

G

0.33/0.44

[7]

Japanese (425/191)

NTG, VCDR

T

0.138/0.223

[8]

rs518394

CDKN2B-AS1

Japanese (740/2723)

POAG, RNFL

C

0.104/0.141

[4]

rs4977756

CDKN2B-AS1

Australian (892/4582)

OAG, advanced OAG

A

0.67/0.60

[1]

Chinese (1157/934)

not associated with POAG

G

0.214/0.227

[2]

Japanese (740/2723)

POAG,VFD, RNFLT

G

0.213/0.258

[4]

African Caribbean 272/165

no association with POAG

G

0.33/0.358

  [3]

rs2157719

CDKN2B-AS1

Chinese (1157/934)

POAG (HTG& NTG),IOP

G

0.092/0.133

[2]

 Saudi Arabian (85/95)

CDR

G

0.182/0.818

[9]

Japanese (565/1104)

IOP

T

0.898/0.838

[10]

rs10120688

CDKN2B-AS1

Japanese (740/2723)

POAG

G

0.306/0.349

[4]

Australian and New Zealand (334/434)

NTG, advanced POAG,VCDR, IOP

A

0.596/0.4856 (NTG),0.5372/0.4856

[11]

rs7049105

CDKN2B-AS1

Chinese (1157/934)

POAG (HTG& NTG)

A

0.319/0.361

[2]

Australian and New Zealand (892/4582)

POAG (HTG& NTG), IOP, VCDR

G

0.5787/0.4609 (NTG), 0.5299/0.4609 (HTG)

[11]

rs523096

CDKN2B-AS1

Japanese (620/578)

NTG

A

0.825/0.904

[12]

Chinese (1157/934)

POAG (HTG& NTG), IOP

C

0.095/0.135

[2]

Abbreviations: NTG- Normal Tension Glaucoma; HTG- High Tension Glaucoma; IOP- Intra Ocular Pressure; POAG- Primary Open-Angle Glaucoma; VCDR- Vertical Cup-Disc Ratio; CDR- Cup to Disc Ratio; RNFLT-Retinal Nerve Layer Thickness

Point 3-Reviewer’s comment: Please be clear about the populations that prior studies have been performed in. The authors should emphasize the lack of diversity in many of the prior studies.

Response: Supplementary Table S2 describes SNPs associated with 9p21 locus in all prior GWAS studies across various populations. Genomic studies demonstrating the association of this locus in African Americans is particularly lacking highlighting lack of diversity of previous studies.

We have modified lines 66-68 in the revised manuscript to read, Published studies have reported the association of the 9p21 locus with POAG mainly in Caucasian, Asian [28] [17] and in a few African populations [29-31] (Supplementary Table S2) underlining lack of diversity in previous studies”

Point 4-Reviewer’s comment: It is important to be aware of and present appropriately the differences and use of race, ethnicity, and ancestry. As genetic ancestry evaluation is not mentioned in this paper for characterization of African Americans, it would be helpful to know if the subjects were self-identified, third-party identified, or something else. 

Response: All the subjects enrolled in the study have self-identified themselves as African American individuals (Black, African decent, or Afro Caribbean) and, aged 35 years or older, from the Philadelphia region (lines 81-83 in the manuscript). Genetic ancestry evaluation was not performed for the case/control individuals in this study but will be performed as a part of an ongoing GWAS study.

Point 5-Reviewer’s comment: Please be clear whether variants in genes of interest have associated with POAG endophenotypes in AAs previously. 

Response: We included an additional table in the revised manuscript, Supplementary Table S2 (also mentioned above). This table shows all GWAS associated variants in the 9p21 locus associated with POAG endophenotypes reported across different populations. To date, only high IOP is found to be associated with variant in CDKN2B-AS1 in African Americans [13]. However, other POAG endophenotypic traits like central corneal thickness (CCT), retinal nerve fiber layer thickness (RNFLT), and cup disc ratio (CDR) are considered risk factors in African Americans [14], [15], but not associated with CDKN2B-AS1 and CDKN2B variants in AAs in previous studies [3,16].

Point 6-Reviewer’s comment: Sections 2.1 and 2.2 mention blood and saliva as sources of DNA; but section 2.2 does not mention any statistical adjustment for these differences in DNA source - please comment. 

Response: We did not perform statistical adjustment for the differences in DNA source. In a previous study from our laboratory, we reported that the genotyping concordance between the saliva and DNA samples obtained from blood is greater than 99.996%[17]. Hence, we did not perform statistical adjustment for these differences in the DNA obtained for our studies.

Point 7-Reviewer’s comment: Section 2.3: With regard to the age of the POAG patients, from which retinal tissues were surgically removed, is there any statistical adjustment made or necessary for the difference in age between tissue sources from cases and controls?

Our Response: Due to limited number of samples for both cases [mean age (40.44±2.3) and controls (mean age 76±4.9, n=3)], we did not perform statistical adjustment. We believe that adjustment may not be necessary for evaluating gene expression differences as the sample size is limited; and the expression of CDKN2A, CDKN2B and CDKN2B-AS1 transcripts in POAG cases is significantly higher than in the controls even when compared to controls with mean age of 76 years.

Point 8-Reviewer’s comment: Section 3.1: last sentence of paragraph 1 - is this consistent with prior knowledge? 

Our Response: Yes, the SNP rs1412831 was not associated with POAG or any quantitative endophenotypic traits in previously published studies. Therefore, our result i.e, “We did not find any significant association of rs1412831 with POAG and quantitative POAG risk factors” is consistent with prior knowledge. However, we have removed references related to rs1412831 including this sentence from the manuscript to avoid confusion and improve the clarity of our results.

Point 9-Reviewer’s comment: Section 3.2: second sentence "These findings were consistent with our results..." - what results is this referring to?

Response: This sentence in Section 3.2 was referring to results in Table.2, where we observed an increased risk for POAG in patients with a homozygous A allele for SNP rs4977756. Since this sentence was pointing to our study results, we chose to remove this to avoid confusion. The revised sentence in lines 250-251 now reads as, “We observed an increased risk for POAG in patients with a homozygous A allele for SNP rs4977756. These findings were consistent with published studies”.

Point 10-Reviewer’s comment: Section 3.3: first sentence - please reword, this is unclear.

Our Response: As suggested by the reviewer, we have modified this sentence to read as, “An increased risk for POAG in patients with SNP rs4977756 and repressor activity of the genomic region surrounding this SNP by luciferase reporter assays suggests that this region may be functionally relevant to glaucoma pathogenesis”,  in lines 264-266 in the manuscript

References:

  1. Burdon, K.P.; Macgregor, S.; Hewitt, A.W.; Sharma, S.; Chidlow, G.; Mills, R.A.; Danoy, P.; Casson, R.; Viswanathan, A.C.; Liu, J.Z., et al. Genome-wide association study identifies susceptibility loci for open angle glaucoma at TMCO1 and CDKN2B-AS1. Nat Genet 2011, 43, 574-578, doi:10.1038/ng.824.
  2. Chen, Y.; Hughes, G.; Chen, X.; Qian, S.; Cao, W.; Wang, L.; Wang, M.; Sun, X. Genetic Variants Associated With Different Risks for High Tension Glaucoma and Normal Tension Glaucoma in a Chinese Population. Invest Ophthalmol Vis Sci 2015, 56, 2595-2600, doi:10.1167/iovs.14-16269.
  3. Cao, D.; Jiao, X.; Liu, X.; Hennis, A.; Leske, M.C.; Nemesure, B.; Hejtmancik, J.F. CDKN2B polymorphism is associated with primary open-angle glaucoma (POAG) in the Afro-Caribbean population of Barbados, West Indies. PLoS One 2012, 7, e39278, doi:10.1371/journal.pone.0039278.
  4. Yoshikawa, M.; Nakanishi, H.; Yamashiro, K.; Miyake, M.; Akagi, T.; Gotoh, N.; Ikeda, H.O.; Suda, K.; Yamada, H.; Hasegawa, T., et al. Association of Glaucoma-Susceptible Genes to Regional Circumpapillary Retinal Nerve Fiber Layer Thickness and Visual Field Defects. Invest Ophthalmol Vis Sci 2017, 58, 2510-2519, doi:10.1167/iovs.16-20797.
  5. Kral, B.G.; Mathias, R.A.; Suktitipat, B.; Ruczinski, I.; Vaidya, D.; Yanek, L.R.; Quyyumi, A.A.; Patel, R.S.; Zafari, A.M.; Vaccarino, V., et al. A common variant in the CDKN2B gene on chromosome 9p21 protects against coronary artery disease in Americans of African ancestry. J Hum Genet 2011, 56, 224-229, doi:10.1038/jhg.2010.171.
  6. Fan, B.J.; Wang, D.Y.; Pasquale, L.R.; Haines, J.L.; Wiggs, J.L. Genetic variants associated with optic nerve vertical cup-to-disc ratio are risk factors for primary open angle glaucoma in a US Caucasian population. Invest Ophthalmol Vis Sci 2011, 52, 1788-1792, doi:10.1167/iovs.10-6339.
  7. Dimasi, D.P.; Burdon, K.P.; Hewitt, A.W.; Fitzgerald, J.; Wang, J.J.; Healey, P.R.; Mitchell, P.; Mackey, D.A.; Craig, J.E. Genetic investigation into the endophenotypic status of central corneal thickness and optic disc parameters in relation to open-angle glaucoma. Am J Ophthalmol 2012, 154, 833-842 e832, doi:10.1016/j.ajo.2012.04.023.
  8. Mabuchi, F.; Sakurada, Y.; Kashiwagi, K.; Yamagata, Z.; Iijima, H.; Tsukahara, S. Association between genetic variants associated with vertical cup-to-disc ratio and phenotypic features of primary open-angle glaucoma. Ophthalmology 2012, 119, 1819-1825, doi:10.1016/j.ophtha.2012.02.044.
  9. Abu-Amero, K.K.; Kondkar, A.A.; Mousa, A.; Almobarak, F.A.; Alawad, A.; Altuwaijri, S.; Sultan, T.; Azad, T.A.; Al-Obeidan, S.A. Analysis of Cyclin-Dependent Kinase Inhibitor-2B rs1063192 Polymorphism in Saudi Patients with Primary Open-Angle Glaucoma. Genet Test Mol Biomarkers 2016, 20, 637-641, doi:10.1089/gtmb.2016.0140.
  10. Shiga, Y.; Nishiguchi, K.M.; Kawai, Y.; Kojima, K.; Sato, K.; Fujita, K.; Takahashi, M.; Omodaka, K.; Araie, M.; Kashiwagi, K., et al. Genetic analysis of Japanese primary open-angle glaucoma patients and clinical characterization of risk alleles near CDKN2B-AS1, SIX6 and GAS7. PLoS One 2017, 12, e0186678, doi:10.1371/journal.pone.0186678.
  11. Burdon, K.P.; Crawford, A.; Casson, R.J.; Hewitt, A.W.; Landers, J.; Danoy, P.; Mackey, D.A.; Mitchell, P.; Healey, P.R.; Craig, J.E. Glaucoma risk alleles at CDKN2B-AS1 are associated with lower intraocular pressure, normal-tension glaucoma, and advanced glaucoma. Ophthalmology 2012, 119, 1539-1545, doi:10.1016/j.ophtha.2012.02.004.
  12. Takamoto, M.; Kaburaki, T.; Mabuchi, A.; Araie, M.; Amano, S.; Aihara, M.; Tomidokoro, A.; Iwase, A.; Mabuchi, F.; Kashiwagi, K., et al. Common variants on chromosome 9p21 are associated with normal tension glaucoma. PLoS One 2012, 7, e40107, doi:10.1371/journal.pone.0040107.
  13. Liu, Y.; Hauser, M.A.; Akafo, S.K.; Qin, X.; Miura, S.; Gibson, J.R.; Wheeler, J.; Gaasterland, D.E.; Challa, P.; Herndon, L.W., et al. Investigation of known genetic risk factors for primary open angle glaucoma in two populations of African ancestry. Invest Ophthalmol Vis Sci 2013, 54, 6248-6254, doi:10.1167/iovs.13-12779.
  14. Girkin, C.A.; Sample, P.A.; Liebmann, J.M.; Jain, S.; Bowd, C.; Becerra, L.M.; Medeiros, F.A.; Racette, L.; Dirkes, K.A.; Weinreb, R.N., et al. African Descent and Glaucoma Evaluation Study (ADAGES): II. Ancestry differences in optic disc, retinal nerve fiber layer, and macular structure in healthy subjects. Arch Ophthalmol 2010, 128, 541-550, doi:10.1001/archophthalmol.2010.49.
  15. La Rosa, F.A.; Gross, R.L.; Orengo-Nania, S. Central corneal thickness of Caucasians and African Americans in glaucomatous and nonglaucomatous populations. Arch Ophthalmol 2001, 119, 23-27.
  16. Williams, S.E.; Carmichael, T.R.; Allingham, R.R.; Hauser, M.; Ramsay, M. The genetics of POAG in black South Africans: a candidate gene association study. Sci Rep 2015, 5, 8378, doi:10.1038/srep08378.
  17. Gudiseva, H.V.; Hansen, M.; Gutierrez, L.; Collins, D.W.; He, J.; Verkuil, L.D.; Danford, I.D.; Sagaser, A.; Bowman, A.S.; Salowe, R., et al. Saliva DNA quality and genotyping efficiency in a predominantly elderly population. BMC Med Genomics 2016, 9, 17, doi:10.1186/s12920-016-0172-y.

Reviewer 2 Report

In this manuscript the authors study the functional significance the CDKN2B-AS1 SNP rs4977756 with primary angle glaucoma and the effect of CDKN2B-AS1 expression on the expression of cyclin dependent kinase inhibitor 2B (CDKN2B) gene which lies within exon 1 of CDKN2B-AS1 and is involved in cell cycle regulation in HEK293 cells and human trabecular meshwork (TM) cells. 

This is an important study which attempts to bridge the molecular genetics with functional analysis in POAG patients and cell culture molecular studies. Although the authors are able to demonstrate that there is an association between the rs4977756 SNP and specific phenotypic changes in POAG patients, the studies do not clearly show that this SNP has a regulatory role on TM cells in either normal healthy TM cells or TM cells in a glaucomatous scenario (elevated TGFB2 levels).   

  1. Why did the authors use HEK293 cells, and not TM cells, in the luciferase assay to demonstrate the effects of the rs4977756 variant? This study would be stronger and more novel if the authors demonstrated that this variant affected gene expression in TM cells. As it stands this study is not relevant to POAG, since it used the wrong cell type.

  1. In the PDF there appears to be a problem with the text in Table 3.

  1. Why did the authors use TGFb1 and not TGFB2 in the study? This is important since TGFB2, not TGFb1, is upregulated in POAG and the two growth factors are not equivalent. They have different requirements for receptors and can potentially trigger different signals. In other words, cells can respond differently to the two growth factors.  They need to demonstrates that TGFb1 behaves like TGFB2 in TM cell cultures or redo the experiments with TGFb2.

  1. The authors use a commercial source of TM cells. The use of commercial sources for human TM has proven to be problematic for the field. These commercial cells tend to be poorly characterized and this is particularly true for this commercial source. The actin structure shown on the company’s website does not shown the novel crosslinked actin network called a CLAN associated with TM cells. Also, the cells exhibit very poor levels of the extracellular matrix protein called fibronectin.

Thus, the authors need to provide additional proof that these cultures are predominantly TM cells.  This can be achieved by providing immunofluorescent images showing that the majority of the cells upregulate myocilin in response to dexamethasone. Showing that the cells form cross-lined actin structures called CLANs in response to dexamethasone is another.  

  1. In this manuscript, the authors make the statement: “Significantly, increased 328 expression of cell adhesion molecules in TM cells (CD44, ICAM1, CTNNA1, CTNND1, LAMA1, 329 LAMC1, SPARC, ITGA3, ITGA5 and ITGA8) after CDKN2B-AS1 siRNA treatment supported the presence of senescent environment in TM cells (Figure 4c).” What evidence do they have that these cell adhesion molecules in particular support cellular senescent in TM cells? Can they provide some supporting references?

Round 2

Reviewer 2 Report

The authors addressed my concerns.